# Predicting the stalking of celebrities from measures of persistent pursuit and threat directed toward celebrities, sensation seeking and celebrity worship

**Maria M. Wong**[1]*, **Lynn E. McCutcheon**[2], **Joshua S. Rodefer**[3], **Kenneth Carter**[4]

1 Department of Psychology, Idaho State University, Pocatello, Idaho, United States of America, 2 North American Journal of Psychology, Winter Garden, Florida, United States of America, 3 Department of Psychology, Mercer University, Macon, Georgia, United States of America, 4 Department of Psychology, Emory University, Atlanta, Georgia, United States of America

* wongmari@isu.edu

**Data Availability Statement:** All relevant data are within the paper and its Supporting Information files.

## Abstract

The stalking of celebrities is a serious issue for thousands of celebrities worldwide who are occasionally confronted by fans who merit the label "fanatic." We administered measures of obnoxious celebrity stalking, celebrity worship, persistent pursuit of celebrities, threat directed toward celebrities, boredom susceptibility, disinhibition, experience seeking, thrill and adventure seeking, relationship styles, and anger to 596 college students from the U.S. A. We developed a model consisting of all but the latter five measures that successfully predicted actual obnoxious stalking behaviors of celebrities. Our results partially replicate earlier research and presents some new findings. Individuals who have personal thoughts about their favorite celebrity frequently, feel compelled to learn more about them, pursue them consistently, threatened to harm them and were prone to boredom were more likely to engage in celebrity stalking. Controlling for these predictors, individuals who admire their favorite celebrity almost exclusively because of their ability to entertain were less likely to engage in celebrity stalking.

## Introduction

Stalking can be defined as the unwanted attention, harassment, or invasion of privacy that threatens or intimidates a person [1]. A large-scale study of stalking suggested that it is far more common than many persons would assume. An estimated 1.7 million people are stalked each year in the United States [2]. The numbers of celebrities that are stalked each year is much smaller, of course, but reaches the level of a societal problem "...when fandom becomes fanaticism in the pursuit of association with the celebrity" (p. 287) [1]. The death of John Lennon, as well as attacks on President Reagan and tennis star Monica Seles, bring to mind the seriousness of celebrity stalking. In one study 25 percent of fans admitted to wanting to be a celebrity's spouse or romantic partner [3]. A study of threatening fan letters revealed that

**Funding:** The authors received no specific funding for this work.

**Competing interests:** The authors have declared that no competing interests exist.

many letter writers indicated that they believed they already had some kind of a personal relationship with the celebrity. Almost half of them (41%) apparently perceived themselves as a friend, acquaintance or advisor to the celebrity [4].

The development of a model that successfully predicts the actual stalking of celebrities is a worthwhile priority. In this article we attempt to predict actual celebrity stalking through a combination of variables that previous research has directly linked to the tendency to condone some celebrity stalking behaviors, and to others that appear to be indirectly linked to celebrity stalking.

The *Obnoxious Fan Activities Scale-18* (OFAS-18) was derived from a list of behaviors that fans sometimes direct toward celebrities [1]. The list consisted of 60 behaviors, but it was reduced by eliminating 'normal' fan activities (e.g., seeking autographs, joining a fan club), ambiguous items (writing to a celebrity may or may not be normal depending on the content), and reducing overlap ("obtaining memorabilia" is similar to "purchasing items associated with the celebrity"). The remaining 18 items went far beyond the range of normal fan behaviors [5]. Every item involved a fan activity that could be construed as annoying or downright dangerous to a celebrity (e.g., "following the celebrity in public," "sending threats or threatening objects," "expressing attraction or sexual interest," and "trespassing on the celebrity's property"). One study found that OFAS-18 scores correlated significantly with a measure of attraction to one's favorite celebrity and two related measures of attitudes condoning celebrity stalking, namely persistent pursuit and threat. In the same study, OFAS-18 scores significantly predicted scores on attitudes condoning the threatening type of celebrity stalking [5].

Over the course of two decades, McCutcheon and colleagues [6–11] measured admiration for celebrities, beginning with the underlying notion that admiration could be best studied by conceptualizing it in terms of degrees of admiration for a favorite celebrity. They created scale items to measure the *extent* to which individuals admired their favorite celebrities. To date more than 90 published articles have used the *Celebrity Attitude Scale (CAS)* in one form or another and these studies confirm its convergent and external validities (for example, see Griffith et al., 2013). Most of these studies found personality variables that correlated with *CAS* scores [12].

There is evidence linking scores on the CAS with a measure of condoning celebrity stalking in others. That is, as admiration for one's favorite celebrity increased, so did the tendency to condone the stalking of celebrities in others [5, 13]. There is also evidence that CAS scores are significantly related to OFAS-18 scores [5]. It is important to note that these findings do not mean that fans and celebrity stalkers are the same. The moderate association between fans and celebrity stalkers indicate that even though there are individuals who are in both categories, there are also individuals who are fans but not celebrity stalkers and vice versa. It is therefore important to identify the factors that help us differentiate between these two constructs.

Attempts to develop and validate an indirect measure of celebrity stalking resulted in the *Obsessive Relational Intrusion and Celebrity Stalking* scale (ORI & CS) [13]. Each of the 11 items presents a brief scenario in which the behavior of a fan is described and the respondent rates the inappropriateness of that behavior. Initial exploratory factor analysis revealed a two-factor solution [14], which was subsequently supported by a confirmatory factor analysis using additional data [13]. Seven items loaded on a factor labeled "persistent pursuit," because of items like one in which a fan is depicted following a celebrity on several occasions. The "threat" factor was so named because it contained items that could potentially harm the celebrity, like one in which a fan writes a letter to a celebrity describing sexual acts that the fan would like to perform on the celebrity. As noted above, ORI & CS scores correlated significantly with OFAS-18 scores in a previous study [5].

There appears to be an element of anger in the behavior of some celebrity stalkers. For example, sending threatening letters to a celebrity or making threats, getting into arguments, damaging her or his valued possessions, filing official complaints, and physically threatening a celebrity all hint at some underlying anger. Both the persistent pursuit and threat subscales of the ORI & CS were found to be correlated with a measure of anger [13], so we included that same measure of anger as a possible predictor of stalking behavior in the present study.

Keinlen [15] and McCann [16] have argued that insecure attachment patterns may lead to stalking either because insecure persons feel socially rejected and are motivated to seek approval from a celebrity, or because they are emotionally distant and wish to retaliate against a perceived wrongdoing. One study found that college students who self-reported as insecure tended to condone celebrity stalking, as measured by the ORI & CS [14]. We used a measure of attachment to see if it would predict stalking behavior in the present study.

Sensation seeking can be defined as ". . .the seeking of varied, novel, complex, and intense sensations and experiences, and the willingness to take physical, social, legal, and financial risks for the sake of such experience" (p. 352) [17]. According to the model developed by Zuckerman, sensation seeking can be conceptualized as consisting of four components: thrill-and-adventure-seeking (TAS), the quest for danger and risk; experience-seeking (ES), a quest for new experiences that challenge the mind and senses; disinhibition (Dis), the ability to be spontaneous and unrestrained; and boredom susceptibility (BS), the inability to tolerate minimal amounts of stimulation [18, 19]. Zuckerman developed several versions of a *Sensation Seeking Scale* (SSS) in response to criticism of early versions of the instrument [19]. The concept of sensation seeking has been extensively researched [20], and numerous studies using one form or another of the SSS have shown considerable reliability and validity, including the brief measure of sensation seeking we used in the present study [21].

There is reason to believe that the stalking of one's favorite celebrity might satisfy a high need for thrill-seeking, inasmuch as actual contact with that celebrity might be both thrilling and somewhat dangerous. For example, there is the risk of arrest if that contact involves lawbreaking. Persons who score high on disinhibition tend to act impulsively, with little consideration for the consequences of their actions. Trespassing on a celebrity's property or invading the celebrity's personal space might be a spontaneous act of devotion, but it can get you in trouble with police or the celebrity's bodyguard. Persons who score high on boredom susceptibility dislike "same old, same old" and get irritated when nothing is going on. One way to relieve boredom is to seek contact with a favorite celebrity, who is likely to be viewed as a person who leads an exciting, eventful life. To date there are few studies that have attempted to determine a relationship between sensation-seeking scores and stalking behaviors. One study found that a low resting heart rate predicted stalking of non-celebrities in males, but not females [22], and another found that impulsive sensation seeking mediated between heart rate and aggression in teenage boys [23]. This evidence suggests that we will find that one or more of the sensation-seeking measures correlates significantly with a measure of celebrity stalking.

To summarize, we used scores from several scales that correlated with measures of either the condoning of celebrity stalking or actual stalking behavior. We predicted that the combination of these scores would significantly predict OFAS-18 as a measure of stalking.

## Materials and methods

### Ethics statement

This study was approved by Emory University Institutional Review Board (exempt), Mercer University Institutional Review Board and Idaho State University Human Subjects Committee. We obtained informed consent from all participants.

## Participants

After we obtained permission from the IRBs of our respective universities, we recruited 620 participants from three universities located in two states: Idaho and Georgia. We excluded 24 participants for failure to provide any responses to a particular scale. This resulted in a final sample size of 596 ($M_{age}$ = 20.22 years, $SD_{age}$ = 3.67). The final sample sizes for each campus were as follows: Idaho ($n$ = 210) and Georgia ($n$ = 188 and 198 at two universities). Our final sample self-identified as 395 females (67%), 194 males (33%), and seven who declined to report a gender. Further, they self-identified as White (51%), Hispanic/Latino/Spanish origin (9%), African American/Black (13%), Asian American/Asian (24%), American Indian (1%), and others (e.g., biracial 2%). Participants completed this study as part of a research participation module or extra credit in a psychology course (ranging from introductory to advanced levels of psychology courses), both accounting for a minimal amount of points in each course. An a priori power analysis using G*Power (Faul et al., 2007) indicated that a total sample size of 123 would be needed to detect a medium effect size of $d > 0.15$ with 11 predictors in a multiple regression model with 80% power and alpha at .05.

## Measures

**Obnoxious Fan Activities Scale-18 (OFAS-18).** OFAS-18 is a reduction from the 60-item *Fan Activities Scale* [1], a list of behaviors that fans sometimes direct toward celebrities. This reduction was accomplished by eliminating 'normal' fan activities (e.g., seeking autographs, going to shows at which the celebrity is appearing), ambiguous items (writing to a celebrity may or may not be normal depending on the content), and reducing overlap ("expressing sexual interest" is similar to "sexually coercing her/him"). The remaining 18 items went far beyond the range of normal fan behaviors. Every item was prefaced by the following question: "Since the age of 16, how often, if at all, have you ever engaged in any of the following activities?" Answers ranged from 1, "never," to 5, "frequently." Sample items include "following the celebrity while out in public," going to or waiting at the celebrity's hotel," "expressing attraction or sexual interest," and "trespassing on the celebrity's property." High scores indicate actual celebrity stalking behavior. In a previous study Cronbach's alpha was .95 [5]. Cronbach's alpha for the OFAS-18 in the current study was .91.

**Obsessional Relational Intrusion and Celebrity Stalking scale (ORI & CS).** ORI & CS is an 11-item Likert-type scale anchored by "very inappropriate" at 1 and "very appropriate" at 7. Factor analysis identified two factors, "persistent pursuit" (7 items; e.g., fan wrote to celeb about enjoying looking at publicity photos of celeb), and "threat," (4 items; e.g., fan made vague warnings that something bad will happen to celeb) Cronbach's alpha was .80 for persistent pursuit, and .77 for threat [13]. In another study [14], ORI & CS was found to be significantly correlated with all three subscales of the CAS, and measures of anger and insecure attachment. Cronbach's alpha for the total ORI & CS in that study was .79. Cronbach's alpha in the current study was .97 for persistent pursuit and .98 for threat.

**Celebrity Attitude Scale (CAS).** CAS has been shown to have good psychometric properties over the course of several studies [6, 8, 9, 11, 24, 25]. It consists of 23 items. The response format for the CAS is a 5-point scale, ranging from 1(*strongly disagree*) to 5(*strongly agree*). The CAS measures three dimensions of celebrity admiration that emerged from factor analysis [10]. The first subscale is entertainment-social (10 items; e.g., "My friends and I like to discuss what my favorite celebrity has done, alpha = .83), intense-personal (9 items; e.g., "I have frequent thoughts about my favorite celebrity, even when I don't want to," alpha = .89), and borderline pathological (4 items; e.g., "I often feel compelled to learn the personal habits of my favorite celebrity;" alpha = .72). High scores on each subscale suggest a person who is strongly

attached to a favorite celebrity. Across several studies total scale Cronbach's alpha values ranged from .84 to .94 [10]. Cronbach's alpha for the CAS in the current study was .89, .87 and .66 for the entertainment-social, intense-personal and borderline pathological subscales respectively. Cronbach's alpha for the whole CAS was .93.

**Brief Sensation Seeking Scale (BSSS).** BSSS [21] consists of eight of the best items from the original scale developed by Zuckerman [26]. Two items are derived from each of the four content domains: experience seeking (ES; "I would like to explore strange places"); boredom susceptibility (BS; "I get restless when I spend too much time alone"); thrill and adventure seeking (TAS; "I would like to do frightening things"); and disinhibition (DIS; "I like wild parties"). Each item is responded to on a Likert scale anchored by "1 = strongly disagree" and "5 = strongly agree." We added four items, one to each of the subscales, in an effort to increase the spread of scores. None of the 12 items are reverse-scored, thus high scores suggest a tendency to be a strong sensation seeker. High scores on the eight-item version predicted tobacco and marijuana use, deviance, poor quality of home life [17], and a borderline pathological addiction to a favorite celebrity [27]. In previous studies the BSSS had Cronbach alphas of .73 [27] and .76 [21]. Cronbach's alpha for the total BSSS-12 in the current study was .77.

**Relationships Questionnaire (RQ).** RQ [28] consists of four brief descriptions of relationship styles, one indicative of a secure attachment to parents, the other three suggesting three different types of insecure attachments. The RQ has been widely used and has been favorably reviewed [29]. We used a modified version described in [14]. Each relationship description is followed by a seven-point, Likert-type scale with "very unlike me" at 1 and "very like me" at 7. For example, the secure description read "It was easy for me to become emotionally close to my parents or guardians in the first 16 years of my life. I was comfortable depending on them and having them depend on me. I didn't worry about being alone or having them not accept me." If the respondent's highest score was on the secure description, we categorized the respondent as "one," securely attached. If the highest score was recorded on any of the other three scales, we categorized the respondent as "zero," insecurely attached. In a previous study, those participants categorized as insecurely attached were significantly more likely to score "high" on the ORI & CS [14].

**Multidimensional Anger Inventory-Brief (MAI-B).** From a lengthy scale designed to measure marital abuse, Dutton [30] developed the MAI-B, based on the three psychometrically best items. The items are rated on a five-point Likert scale, anchored by 1, "completely undescriptive of you," and 5, "completely descriptive of you." One of the three items is "I get so angry, I feel that I might lose control." High scores suggest greater trait anger. Cronbach's alpha was .72 in one study [31], and .88 in the present study.

## Procedure

This study was conducted online. After obtaining IRB approval on each campus and informed consent, the participants completed a survey containing the measures described above online. Presentation of scales were randomized to minimize the probability of a systematic order effect. Participants were briefed and thanked for their participation after their session was complete.

## Results

### Descriptive statistics

Means, standard deviations and the possible range of scores for all variables were presented in Table 1. Mean scores for each scale are consistent with findings reported in previous studies.

**Table 1. Means, standard deviations, and possible range of scores for each measure.**

|  | Mean (SD) | Possible range of scores |
|---|---|---|
| Obnoxious Fan AS-18 | 19.24 (3.80) | 18–90 |
| CAS Entertainment Social | 28.03 (7.95) | 10–50 |
| CAS Intense Personal | 15.90 (6.32) | 9–45 |
| CAS Borderline Pathological | 7.98 (2.90) | 4–20 |
| ORI & CS Persistent Pursuit | 15.40 (6.56) | 7–49 |
| ORI & CS Threat | 5.93 (3.45) | 4–28 |
| BSSS Experience Seeking | 10.52 (2.28) | 3–15 |
| BSSS Boredom Susceptibility | 8.24 (2.14) | 3–15 |
| BSSS Thrill & Adventure Seeking | 9.18 (3.13) | 3–15 |
| BSSS Disinhibition | 7.92 (2.62) | 3–15 |
| RQ (attachment) | 44% secure (Categorical) | 0–1 |
| MAI-B (anger) | 6.13 (2.83) | 3–15 |

Zero-order correlations for all major variables were presented in Table 2. The dependent variable OFAS-18 was correlated significantly with the three subscales of CAS as well as with both subscales of ORI & CS. It was also correlated with BSSS Boredom susceptibility and Disinhibition. However, it was not correlated with BSSS Thrill and adventure seeking, BSSS Experience seeking, RQ or the MAI.

## Predictors of celebrity stalking

Multiple linear regression models were used to examine the relationships between the above measures and OFAS-18. Sex, age and race (0 = non-white and 1 = white) were controlled for in all analyses. These demographic variables were not significant in any models. They were removed from the final models presented here. The results showed that the three subscales of CAS, the two subscales of ORI & CS and BSSS Boredom susceptibility uniquely and

**Table 2. Zero-order correlations for all measures.**

|  | OFAS-18 | CAS ES | CAS IP | CAS BP | ORI & CS PP | ORI & CAS T | BSS ES | BSS BS | BSS TAS | BSS Dis | RQ | MAI-B |
|---|---|---|---|---|---|---|---|---|---|---|---|---|
| OFAS-18 | – | | | | | | | | | | | |
| CAS ES | .14 | – | | | | | | | | | | |
| CAS IP | .31 | .68 | – | | | | | | | | | |
| CAS BP | .28 | .72 | .71 | – | | | | | | | | |
| ORI & CS PP | .25 | .14 | .26 | .26 | – | | | | | | | |
| ORI & CS T | .37 | .04 | .21 | .21 | .76 | – | | | | | | |
| BSSS ES | -.01 | .10 | .01 | .01 | .00 | -.05 | – | | | | | |
| BSSS BS | .10 | .10 | .04 | .04 | .10 | .04 | .40 | – | | | | |
| BSSS TAS | -.04 | .00 | -.05 | .09 | .05 | -.03 | .47 | .35 | – | | | |
| BSSS DIS | .12 | .10 | .13 | .13 | .07 | .04 | .40 | .39 | .41 | – | | |
| RQ | -.07 | .00 | -.02 | -.02 | -.03 | -.07 | -.11 | -.10 | -.14 | -.12 | – | |
| MAI-B | .03 | .15 | .15 | .15 | .06 | -.03 | .15 | .15 | .07 | .24 | -.11 | – |

Note. OFAS-18 = Obnoxious Fan AS-18; CAS ES = CAS Entertainment Social; CAS IP = CAS = Intense Personal; CAS BP = CAS Borderline Pathological; ORI & CS PP = ORI & CS Persistent Pursuit; ORI & CS T = ORI & CS Threat; BSSS ES = BSSS Experience Seeking; BSSS BS = BSSS Boredom Susceptibility; BSSS TAS = BSSS Thrill & Adventure Seeking; BSSS DIS = BSSS Disinhibition; RQ = Relationship Questionnaire; MAI-B = Multidimensional Anger Inventory-Brief. All values of plus or minus .09 or higher are significant at $p < .05$; values of plus or minus .11 or higher are significant at $p < .01$; values of plus or minus .14 or higher are significant at $p < .001$. Correlation between RQ (a dichotomous variable) and OFAS-18 is a point-biserial correlation.

**Table 3. Multiple linear regression using CAS, ORI & CS, BSSS, RQ and MAI-B to predict OFAS-18.**

|  | B (SE) | 95% CI of B | β | 95% CI of β | t | p |
|---|---|---|---|---|---|---|
| CAS ES | -.07 (.03) | -.12 to -.01 | -.14 | -.27 to -.03 | -2.46 | .02 |
| CAS IP | .14 (.03) | .07 to .20 | .23 | .12 to .34 | 4.02 | .00 |
| CAS BP | .23 (.08) | .07 to .39 | .17 | .05 to .30 | 2.73 | .01 |
| ORI & CS PP | -.08 (.03) | -.14 to -.01 | -.13 | -.25 to -.02 | -2.31 | .02 |
| ORI & CS T | .42 (.06) | .30 to .55 | .38 | .28 to .51 | 6.63 | .00 |
| BSSS ES | -.04 (.07) | -.18 to .11 | -.02 | -.11 to .07 | -.54 | .63 |
| BSSS BS | .15 (.07) | .00 to .29 | .08 | .00 to .17 | 1.97 | .05 |
| BSSS TAS | -.09 (.05) | -.05 to .21 | -.07 | -.16 to .02 | -1.59 | .11 |
| BSSS Dis | .08 (.07) | -.19 to .02 | .05 | -.04 to .15 | 1.18 | .24 |
| RQ | -.20 (.28) | -.76 to .36 | -.03 | -.10 to .05 | -.69 | .49 |
| MAI-B | -.01 (.05) | -.11 to .10 | -.01 | -.08 to .07 | -.16 | .92 |

significantly predicted OFAS-18. The model significantly explained about 22.4% of the variance of OFAS-18, $F(11,587) = 15.42$, $p < .001$, $R^2 = .224$. The unstandardized and standardized beta coefficients, the $t$ and $p$ values were presented in Table 3. We entered the variables simultaneously for this analysis. We also conducted several stepwise multiple regression models and found that the same six variables significantly predicted scores on OFAS-18.

The dependent variable OFAS-18 is positively skewed. About 64% of participants reported that they did not engage in any celebrity stalking behaviors and most participants had a lower score. We therefore supplemented the main analyses by conducting Poisson regression, a type of generalized linear models designed for skewed event data [32, 33]. The results showed that CAS Intense Personal (*Wald $\chi^2(1)$* = 8.61, $p = .003$), CAS Borderline pathological (*Wald $\chi^2(1)$* = 3.97, $p = .046$) and ORI & CS Threat were significant predictors (*Wald $\chi^2(1)$* = 22.09, $p = .000$). CAS Entertainment Social was marginally significant (*Wald $\chi^2(1)$* = 3.27, $p = .07$). These predictors are all significant in the multiple regression models. ORI & CS Persistent Pursuit (*Wald $\chi^2(1)$* = 2.52, $p = .11$) and BSSS Boredom Susceptibility (*Wald $\chi^2(1)$* = 2.46, $p = .12$) has a p-value of about .1. Though not statistically significant in the Poisson regression model, their p-values were lower than all of the remaining predictors. Thus, the results of the Poisson regression model are largely consistent with the multiple regression analyses.

The fact that participants came from three different universities raised the question of whether multi-level regression was needed to analyze the data. To address this issue, we calculated the intra-class correlation (ICC) to determine the degree of variability in the dependent variable that could be attributed to schools (clusters). The ICC were estimated to be .02, meaning that schools explained roughly 2% of the variability in OFAS-18. An ICC value of .05 or higher is typically used as the cut-off value to determine whether multi-level analyses are necessary [34]. Our data show that there is no significant clustering effect attributable to schools. The single-level regression analyses presented in the paper are therefore appropriate.

## Discussion

The main objective of this study was to assess the association between OFAS-18, a celebrity stalking measure, with measures of celebrity worship, persistent pursuit of celebrities, threat directed toward celebrities, boredom susceptibility, disinhibition, experience seeking, thrill and adventure seeking and relationship styles. The mean scores we obtained on our study variables were consistent with mean scores found in previous studies using the CAS, OFAS-18 and ORI & CS. For example, the total CAS score of 51.91 is similar to the total CAS mean score (57.74) obtained by McCutcheon et al. [7], and our mean scores for ORI & CS Persistent

pursuit (15.40) and ORI & CS Threat (5.93) are consistent with the mean scores (17.97 & 6.20 respectively) obtained by McCutcheon, Aruguete et al. [5], and the mean scores (17.55 & 4.75 respectively) obtained by McCutcheon et al. [13]. Our mean scores are not directly comparable to those obtained by Hoyle et al. [21] and Lopez-Bonilla and Lopez-Bonilla (2010) on the BSSS because they used the eight-item version, but both of their studies found that mean scores were highest on the Experience Seeking subscale, just as our study did. The consistency over time and over many studies, combined with high Cronbach alphas, allows us to place considerable faith in our results. We should note, however, that adding one item to each of the four subscales of the BSSS did little to raise its internal reliability value.

We hypothesized that a combination of scale scores that had been previously associated with either the condoning of celebrity stalking in others or actual celebrity stalking, along with four sensation seeking scales, would successfully predict celebrity stalking in the present study. Our hypothesis was partially confirmed, although scores on anger (MAI-B), BSSS Experience seeking, BSSS thrill and adventure seeking, and attachment (RQ) contributed virtually nothing to our predictive model.

Anger and attachment did not predict OFAS-18 scores. Previous research linking anger with OFAS-18 scores was only indirect and weak. That is, anger correlated weakly with a measure of stalking in general in one of the 2006 studies [13], but was not included as a variable in the other celebrity stalking study from the same year [14], nor was it included in the 2016 study [5]. Our measure of attachment correlated weakly, though significantly ($r$ = -.16) with ORI & CS in one of the 2006 studies [14]. The current study found non-significant associations between celebrity stalking (as measured by OFAS-18) and both anger and attachment. This is the first time that these relations were reported in the literature. Previous studies did not use OFAS-18 to measure celebrity stalking. Taken as a whole, both current and past findings indicate that celebrity stalking is either uncorrelated or weakly correlated with anger or attachment.

It is worth noting that the zero-order correlation between CAS Entertainment-social and our celebrity stalking measure was somewhat lower (though in the same direction) than the correlations between CAS Intense-personal and Borderline pathological and our celebrity stalking measure. In the multiple regression analyses, the relationship between CAS entertainment-social is opposite in direction from the other predictors of obnoxious celebrity stalking. In other words, after controlling for other variables in the analyses, participants who admired their favorite celebrity almost exclusively because of their ability to entertain, were significantly *less* likely to engage in obnoxious celebrity stalking. This finding is consistent with a body of research and the absorption-addiction model showing that individuals who admire their favorite celebrity only for entertainment and social reasons are likely to be more mentally healthy than those who become addicted [35–37]. In contrast, both CAS intense-personal and CAS borderline-pathological positively predict OFAS-18. Those who have personal thoughts about their favorite celerity frequently and feel compelled to learn more about them are more likely to engage in celebrity stalking.

We found that the best predictors of obnoxious celebrity stalking behavior from previous studies also successfully predicted obnoxious celebrity stalking in the present study. In addition to CAS Intense-personal, CAS Borderline-pathological, ORI & CS persistent-pursuit, ORI threat and BSSS boredom-susceptibility are also associated with OFAS-18. Those who pursue their favorite celebrity persistently and threaten to harm them were more likely to stalk them. In general, our four measures of sensation seeking failed to predict obnoxious celebrity stalking behavior. The one exception, boredom susceptibility, was the weakest of the six significant predictors of obnoxious celebrity stalking behavior.Our finding is consistent with previous research. Boredom proneness correlated with all three subscales of the CAS in two studies [38,

39], and Maltby et al. [35] found that excitement seeking correlated positively with the entertainment/social subscale of the CAS.

It is worth noting that RQ, a secure attachment measure, was not significantly correlated with OFAS-18 scores. It was also a non-significant predictor in all multiple regression analyses. Based on our findings, insecure attachment does not predict celebrity stalking. This is inconsistent with previous empirical studies and theories [14–16]. The inconsistency may have stemmed from the different measures used to measure attachment in past research. Alternatively, the relationship between attachment and obnoxious fan activities may be moderated by other variables such as social isolation or loneliness. These are questions that could be investigated by future research.

Our study is not without some limitations. As with any correlational research, it is impossible to determine causation. Furthermore, our research relied on self-report data. Also noteworthy is the fact that we cannot generalize our findings to the general population. Table 1 shows that the vast majority of our sample scored rather low on the OFAS-18, our measure of celebrity stalking behavior. It remains to be seen whether research conducted on a sample of more serious celebrity stalkers will yield findings similar to the one reported in this study. Another limitation is that our data were collected from Idaho and Georgia. Perhaps data collected from Hollywood would yield higher scores on OFAS-18.

## Supporting information

**S1 Data.**
(ZIP)

**S1 File.**
(DOC)

## Author Contributions

**Conceptualization:** Maria M. Wong, Lynn E. McCutcheon, Joshua S. Rodefer, Kenneth Carter.

**Data curation:** Maria M. Wong, Joshua S. Rodefer, Kenneth Carter.

**Formal analysis:** Maria M. Wong.

**Investigation:** Maria M. Wong, Lynn E. McCutcheon, Joshua S. Rodefer, Kenneth Carter.

**Methodology:** Maria M. Wong, Lynn E. McCutcheon, Joshua S. Rodefer, Kenneth Carter.

**Project administration:** Maria M. Wong, Lynn E. McCutcheon, Joshua S. Rodefer, Kenneth Carter.

**Resources:** Maria M. Wong, Kenneth Carter.

**Writing – original draft:** Maria M. Wong, Lynn E. McCutcheon, Joshua S. Rodefer, Kenneth Carter.

**Writing – review & editing:** Maria M. Wong, Lynn E. McCutcheon, Joshua S. Rodefer, Kenneth Carter.

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
