## [Decision Letter · Decision Letter 0]

22 Jun 2022

PONE-D-22-09764Predicting the Stalking of Celebrities from Measures of Persistent Pursuit and Threat Directed toward Celebrities, Sensation Seeking and Celebrity WorshipPLOS ONE

Dear Dr. Wong,

Thank you for submitting your manuscript to PLOS ONE. After careful consideration, we feel that it has merit but does not fully meet PLOS ONE’s publication criteria as it currently stands. Therefore, we invite you to submit a revised version of the manuscript that addresses the points raised during the review process.

I have been fortunate to receive 2 detailed reviews of the submission; please let me take this opportunity to thank the reviewers for their diligence. I would be grateful if you would consider all the points made by the reviewers and to either address these with revisions to the manuscript or explain why you have chosen not to make the recommended changes in your rebuttal letter.

We look forward to receiving your revised manuscript.

Kind regards,

Richard Rowe

Academic Editor

PLOS ONE

Journal Requirements:

Reviewers' comments:

Reviewer's Responses to Questions

**Comments to the Author**

1. Is the manuscript technically sound, and do the data support the conclusions?

Reviewer #1: Yes

Reviewer #2: No

2. Has the statistical analysis been performed appropriately and rigorously? 

Reviewer #1: Yes

Reviewer #2: No

3. Have the authors made all data underlying the findings in their manuscript fully available?

Reviewer #1: No

Reviewer #2: Yes

4. Is the manuscript presented in an intelligible fashion and written in standard English?

Reviewer #1: Yes

Reviewer #2: Yes

5. Review Comments to the Author

Reviewer #1: There is a large zip file at the end of the article that I was unable to access or open. Some of the questions I have might be answered there. But the fact remains that the file couldn't be opened after quite a long time of trying.

First point: "In other words, after controlling for other variables in the analyses, participants who admire their favorite celebrity almost exclusively because of their ability to entertain, were significantly less likely to engage in obnoxious celebrity stalking." This point has been made in other publications, the fact that being a fan and all of this other negative behavior are not related at all. In fact, in this case, being a "normal" fan makes it LESS likely that the person would be a stalker. Given the ways in which average fans are stigmatized in our culture, it seems important that this statement not be buried at the end of the article. It should be stated in the abstract, as it is an important finding.

The 18 items on the "obnoxious fan behavior" scale should be included in the article as they are focal and centrai to the findings. Have the authors validated this scale by submitting it to a panel of celebrities and seeing what percentage of them agree with the items? Defining a behavior as obnoxious in the absence of such a procedure seems questionable at best. For example, if you asked a celebrity if they found a "fan" expressing the opinion that they are attractive as obnoxious, how many of them would agree with that? In my own experience and observations, a large percentage of celebrities would simply be flattered. Also the situation under which this might happen would make a huge difference in how it might be received.

"Further research should be conducted on a sample of more serious celebrity stalkers, a sample that presumably would score much higher on our OFAS-18 measure." This statement concerns me. If the current instrument is an early attempt at a way to identify celebrity stalkers, then how would a sample of "more serious celebrity stalkers" be obtained? The reasoning here feels somewhat circular to me.

Overall this is a good study, with solid data and analysis. I like it. However, more care needs to be taken to make clear that the construct "fan" and the construct "celebrity stalker" are completely different and unrelated. The stalkers identified in the article (Chapman, Hinkley) were not "fans" in the truest sense of the word and many others are not as well (although certainly a few might be). An attempt to find stalkers among "fans" seems doomed to failure. And the idea that this could happen contributes to the stigma of being a typical media fan.

Reviewer #2: Thank you for the opportunity to review manuscript # PONE-D-22-09764, titled “Predicting the Stalking of Celebrities from Measures of Persistent Pursuit and Threat Directed toward Celebrities, Sensation Seeking and Celebrity Worship”. Using a large sample of undergraduate and graduate students across three university campuses, the present study examined associations between among a variety of self-report scales, including obnoxious celebrity stalking, celebrity worship, persistent pursuit of celebrities, threat directed toward celebrities, boredom susceptibility, disinhibition, experience seeking, thrill and adventure seeking, relationship styles, and anger. Zero-order correlations and multiple linear regression were used to conduct inferential analyses. I hope the author finds the following comments and questions helpful.

Note that Cohen’s d is transformed to Pearson’s correlation coefficient using the following equation: r = d/sqrt(d^2+4) (written in R code). So, d = .15 transforms to r = .07478995 and R2 = .005593536. The author reports that d = .15 “corresponds to an R2 of approximately .13” and cite Cohen (1988), but this is incorrect, as shown above. Infamously, despite his genius, Cohen (1988) got the transformations between common effect sizes wrong. Please correct the reported R2 and omit the citation to Cohen (1988), as it may be the only time he got something wrong.

As measurement error is bound to vary from study to study, I see no value in reporting “In a previous study Cronbach’s alpha was .95” (line 154).

Related, the author reports Cronbach's alpha for self-report scales, but many have argued that Cronbach’s alpha is not a valid measure of internal consistency (McDonald, 1999; Revelle & Zinbarg, 2009; Sijtsma, 2009; Zinbarg et al., 2005). What some may consider the “actual” reliability scores may be higher or lower when obtained using McDonald’s omega index. It can’t hurt to follow the cited authors and report alpha, omega total, and omega hierarchical. The omega indices can be easily calculated using the ‘psych’ package in R.

More descriptive statistics should be reported for self-report scales, including skew and kurtosis.

Based on the mean (19.24) standard deviation (3.80) and possible range of scores (18-90), it is clear that the Obnoxious Fan Activities Scale was positively skewed and likely zero-inflated. This is problematic because it is the dependent variable in multiple linear regression, which almost guarantees that the residual errors are not normally distributed, and ordinary least squares regression is highly sensitive to deviation from normality. Thus, the regression coefficients, standard errors, and p-values from the multiple linear regression cannot be trusted. The author should think carefully about how to deal with the positive skew of the dependent variable. Many different types of robust regression methods have been developed for this purpose.

On lines 153-153, the author writes “High scores indicate actual celebrity stalking behavior”, referring the Obnoxious Fan Activities Scale. However, because data were collected exclusively from students living in Idaho and Georgia, I doubt seriously that many opportunities were presented to the study participants to follow a celebrity while out in public, go to or wait at a celebrity’s hotel, or trespass on celebrity’s property. Given that such stalking behavior is the focal study variable of interest, did the author sample from the appropriate geographic regions? I would be more convinced in the veracity of student’s self-reports on this scale if data had been collected in LA or NYC.

Related, were participants asked to report the name of the celebrity they claimed to stalk? How much actual celebrity stalking was measured in the current study? Again, based on the few descriptive statistics that were reported, it appears very little.

Regarding the ORI & CS, the author writes “factor analysis identified two factors” without reporting any information on the factor analysis that was allegedly performed. This is very strange. More details need to be reported on the factor analysis that was performed. What factor extraction techniques were utilized? Sequential model tests, the Hull Method, the Empirical Kaiser Criterion, traditional and revised parallel analysis? What estimator was used? What type of factor rotation was used? Factor loadings should be reported. What was the correlation between the two factors, that is, assuming an oblique rotation. Please report on all of the above.

Again, I’m open to be convinced otherwise but I see absolutely no value in reporting Cronbach’s alpha from another study. The author does this for all self-report scales.

The author’s report that “Cronbach’s alpha in the current study for persistent pursuit was .97 and .98 for threat”, which is troubling. Such high Cronbach’s alphas usually indicate redundancy in item content and/or failure of a measure to capture the full breadth of content space, which is antithetical to high internal validity.

What does the author mean by, “good psychometric properties” (line 166)? Please elaborate.

The author makes eye-ball average comparisons between the scales in the current study with previous studies (lines 221-230), which is bizarre. The whole point of inferential statistics is to obviate this type of imprecise, wishy-washy, eye-ball comparison. Has the author never heard of an independent samples t-test? This could be calculated without access to the raw data from previous studies. Technically, given that many of the variables were not normally distributed, the author should use a Mann-Whitney U test to determine if the distributions of scores varied across studies, although this would require access to the previously published data.

The author should consider making better use of the upper off-diagonal cells of Table 2. The author could report either non-parametric correlations (e.g., Kendall’s tau) or partial correlations controlling for demographic variables (age, gender, and race/ethnicity).

It would be good to report 95% confidence intervals for standardized regression coefficients in Table 3.

It would also be good to report exact p-values in Table 3, instead of p-value cut-offs using asterisks.

Note, most researchers no longer favor a backward elimination procedure because all of the automated, sequential, model comparisons have a high likelihood of capitalizing on sampling variability, increasing the likelihood of type-I error.

There was considerably heterogeneity in terms of demographic characteristics in the current study. Age, sex, and race/ethnicity should be included in the multiple linear regression as covariates to ensure the documented associations are not confounded by these variables. This is generally standard practice in studies with non-experimental correlational designs.

There is a clustered pattern to the observations in the present study. Schools were nested within different states (Idaho and Georgia), classrooms were presumably nested within different schools, and participants were nested within classrooms. This clustered or nested structure introduces potential non-independence among residuals, which is another violation of an underlying assumption of multiple linear regression, which renders the results untrustworthy. The author either needs to conduct multi-level modeling or specify a cluster variable and implement a sandwich estimator to account for the non-independence of observations, otherwise the standard errors of regression coefficients will be biased.

The interpretation of results is problematic. For example, the author writes “The fact that mean scores we obtained on our study variables were consistent with mean scores found in previous studies suggests that our participants showed diligence and conscientiousness in responding to our scales. This, combined with high Cronbach alphas, allows us to place considerable faith in our results.” First, as previously noted, many have argued that Cronbach’s alpha is not a valid measure of internal consistency. Moreover, the above interpretation would only sound if (1) participants showed diligence and conscientiousness in responding in previous studies, for which no evidence was provided, (2) there were neither measured nor unmeasured confounders that differed across studies, and (3) the samples in previous studies were drawn from the same population as the samples in the current study. Consequently, the quoted interpretation is tenuous at best.

Other sections of the discussion are equally problematic. For example, the author writes “Our measure of attachment correlated weakly (-.16) with ORI & CS in one of the 2006 studies (McCutcheon, Scott, Jr. 2006) in part because of a large sample size (n = 299).” However, a large sample size is not an explanation for a weak correlation. A large sample size is an explanation for why a weak correlation would be statistically significant at p < .05, but it’s not an explanation for the small point estimate (i.e., not an explanation for the weak correlation itself).

In regard to “Why did anger and attachment not predict OFAS-18 scores?” (line 268). Did the author considered multicollinearity among independent variables and consequent inflation of standard errors? This could be probed empirically by calculating variance inflation factors for the predictors in the multiple regression.

The limitations that the author noted in the last paragraph of the discussion, although correct, further dampen my enthusiasm for the current study. Mainly, the present study was a cross-sectional, non-experimental design that relied solely on self-report scales that were completed by psychology students, which weakens both internal and external validity.

I don't believe the present study warrants publication in PLOS ONE. Nevertheless, I hope the above comments and questions are helpful. Best wishes and good luck.

References:

McDonald, R. P. (1999). Test theory: A unified treatment. Test theory a unified treatment. Hillsdale, NJ: Lawrence Erlbaum. doi:10.4324/9781410601087

Revelle, W., & Zinbarg, R. E. (2009). Coefficients alpha, beta, omega, and the glb: Comments on Sijtsma. Psychometrika, 74(1), 145-154.

Sijtsma, K. (2009). On the use, the misuse, and the very limited usefulness of Cronbach's Alpha. Psychometrika, 74(1), 107–120. doi:10.1007/s11336-008-9101-0

Zinbarg, R. E., Revelle, W., Yovel, I., & Li, W. (2005). Cronbach’s α, Revelle’s β, and McDonald’s ω H: Their relations with each other and two alternative conceptualizations of reliability. Psychometrika, 70(1), 123-133.

6. PLOS authors have the option to publish the peer review history of their article (what does this mean?). If published, this will include your full peer review and any attached files.

Reviewer #1: No

Reviewer #2: No

---

## [Author Response · Author response to Decision Letter 0]

29 Oct 2022

Please see the uploaded response to editor and reviewers document.

---

## [Editor Report · Decision Letter 1]

6 Nov 2022

PONE-D-22-09764R1Predicting the Stalking of Celebrities from Measures of Persistent Pursuit and Threat Directed toward Celebrities, Sensation Seeking and Celebrity WorshipPLOS ONE

Dear Dr. Wong,

Thank you for submitting your manuscript to PLOS ONE. After careful consideration, we feel that it has merit but does not fully meet PLOS ONE’s publication criteria as it currently stands. Therefore, we invite you to submit a revised version of the manuscript that addresses the points raised during the review process.

 Many thanks for your attention to the reviewers' comments. I have decided against sending your paper back to the reviewers at this stage, but I identify a number of issues that I would like you to address before I can make a final decision on this submission. In relation to Reviewer 2's comments: Point 13. Please include details of the Poisson regression modelling in your manuscript, including the rationale for conducting it and the extent to which the findings are similar to your current modelling. Point 16. Please add a few more details on the factor structure identified in the cited paper. Point 22. I agree with the reviewer that the CIs on the standardised betas will be most helpful. The CIs there are as informative regarding significance as the CIs on the b coefficients. Point 26. The reviewer's point is about correlations between observations from the same school; non-independent observations will reduce artificially reduce the variance in the Sample. This problem exists independently from mean level differences differences between schools. I think the reviewer’s suggestion of a clustered regression is a good one. So I would be grateful if you could run these additional analyses which I think are commonly available in most stats packages. 28. I agree with the reviewer that a small correlation does not result from a large sample size. Instead a large sample size provides the power to detect smaller correlations. So the text needs to be re-written to say that there may only be a small correlation between these measures in the population and therefore a large sample is required to identify it as signficant. In my own reading of the revised manuscript:Line 27. Celebrity not “celerity" Line 39. My understanding is that Princess Diana died in a car crash chased by the press rather than stalkers, so I recommend deleting this example. Line 62. Please remove mention of the statistical test used and just focus on the substantive result. Line 132. Again concentrate on the substantive point that your predictors will independently correlate with stalking rather than mentioning multiple regression. Line 240 Head this section “Descriptive Statistics” Line 241-250 Move this comparison to previous studies to the discussion. Line 253 Head this section “Predictors of stalking” or something similar that conveys the contents of the section more clearly. Line 265. Clarify that the control of sex etc was in preliminary models. Line 267 insert “removed”

We look forward to receiving your revised manuscript.

Kind regards,

Richard Rowe

Academic Editor

PLOS ONE
---

## [Author Response · Author response to Decision Letter 1]

4 Jan 2023

The response letter file was uploaded.

---

## [Editor Report · Decision Letter 2]

16 Jan 2023

PONE-D-22-09764R2Predicting the Stalking of Celebrities from Measures of Persistent Pursuit and Threat Directed toward Celebrities, Sensation Seeking and Celebrity WorshipPLOS ONE

Dear Dr. Wong,

Thank you for submitting your manuscript to PLOS ONE. After careful consideration, we feel that it has merit but does not fully meet PLOS ONE’s publication criteria as it currently stands. Therefore, we invite you to submit a revised version of the manuscript that addresses the points raised during the review process.

Very many thanks for carefully addressing the points raised after the previous review. In my view, the paper is ready for publication after a some very minor amendments, as specified here: You have dealt with the multi-level issue raised well. I would be grateful if you could include reference to your analysis testing the necessity for a multi-level approach somewhere in the manuscript. There is no need to provide the details, but please demonstrate you have considered the issue which may come to the minds of many readers. Line 245 are not is TAble 2. Abbreviations need to be defined in the Note Line 272 insert “in” Line 290 delete unnecessary space.==============================

We look forward to receiving your revised manuscript.

Kind regards,

Richard Rowe

Academic Editor

PLOS ONE
---

## [Author Response · Author response to Decision Letter 2]

23 Jan 2023

Please see the uploaded response to editor and reviewers document.

---

## [Editor Report · Decision Letter 3]

26 Jan 2023

Predicting the Stalking of Celebrities from Measures of Persistent Pursuit and Threat Directed toward Celebrities, Sensation Seeking and Celebrity Worship

PONE-D-22-09764R3

Dear Dr. Wong,

We’re pleased to inform you that your manuscript has been judged scientifically suitable for publication and will be formally accepted for publication once it meets all outstanding technical requirements.

Kind regards,

Richard Rowe

Academic Editor

PLOS ONE
---

## [Editor Report · Acceptance letter]

2 Feb 2023

PONE-D-22-09764R3 

Predicting the Stalking of Celebrities from Measures of Persistent Pursuit and Threat Directed toward Celebrities, Sensation Seeking and Celebrity Worship 

Dear Dr. Wong:

I'm pleased to inform you that your manuscript has been deemed suitable for publication in PLOS ONE. Congratulations! Your manuscript is now with our production department. 

Kind regards, 

on behalf of

Professor Richard Rowe 

Academic Editor

PLOS ONE